# AuthFace: Towards Authentic Blind Face Restoration with Face-oriented Generative Diffusion Prior

## Abstract

Blind face restoration (BFR) is a fundamental and challenging problem in computer vision. To faithfully restore high-quality (HQ) photos from poor-quality ones, recent research endeavors predominantly rely on facial image priors from the powerful pretrained text-to-image (T2I) diffusion models. However, such priors often lead to the incorrect generation of non-facial features and insufficient facial details, thus rendering them less practical for real-world applications. In this paper, we propose a novel framework, namely **AuthFace** that achieves highly authentic face restoration results by exploring a face-oriented generative diffusion prior. To learn such a prior, we first collect a dataset of *1.5K* high-quality images, with resolutions exceeding 8K, captured by professional photographers. Based on the dataset, we then introduce a novel face-oriented restoration-tuning pipeline that fine-tunes a pretrained T2I model. Identifying key criteria of quality-first and photography-guided annotation, we involve the retouching and reviewing process under the guidance of photographers for high-quality images that show rich facial features. The photography-guided annotation system fully explores the potential of these high-quality photographic images. In this way, the potent natural image priors from pretrained T2I diffusion models can be subtly harnessed, specifically enhancing their capability in facial detail restoration. Moreover, to minimize artifacts in critical facial areas, such as eyes and mouth, we propose a time-aware latent facial feature loss to learn the authentic face restoration process. Extensive experiments on the synthetic and real-world BFR datasets demonstrate the superiority of our approach. *Codes and datasets will be available upon acceptance.*

## 1 Introduction

Face images captured in natural settings often exhibit various forms of degradation, including compression, blur, and noise (Wang et al., 2021; 2023c). Capturing high-quality (HQ) face images is crucial, as humans are highly sensitive to subtle facial details. Blind face restoration (BFR) aims to reconstruct HQ images from degraded inputs and has rapidly progressed in recent years due to significant research interest. However, BFR remains an ill-posed problem due to the unknown degradation and the loss of valuable information resulting from these complex conditions (Zhou et al., 2022).

Sufficient prior information is critical for HQ reconstruction. Researchers have used geometric and reference priors from sources like (Bulat & Tzimiropoulos, 2018; Kim et al., 2019; Chen et al., 2021; Shen et al., 2018; Yang et al., 2020; Yu et al., 2018; Hu et al., 2020; Zhu et al., 2022; Ren et al., 2019; Dogan et al., 2019; Li et al., 2020a;b; 2018; Chen et al., 2018; Ma et al., 2020) to guide face restoration. These priors, however, are limited by their sensitivity to degradation and inability to capture fine facial details, and can even result in corrupted texture details due to incorrect prior information (Lu et al., 2021). With advancements in generative models, such as StyleGAN (Karras et al., 2020) and VQVAE (Razavi et al., 2019), recent works (Chen et al., 2021; Wang et al., 2021; Chan et al., 2022; Xie et al., 2023; Wang et al., 2022a;b; Zhou et al., 2022; Tsai et al., 2023) have leveraged pretrained networks to derive facial priors, achieving superior results compared to earlier methods. Nonetheless, these approaches still face significant performance declines in handling unseen cases. Denoising diffusion probabilistic models (DDPMs) (Ho et al., 2020) have shown

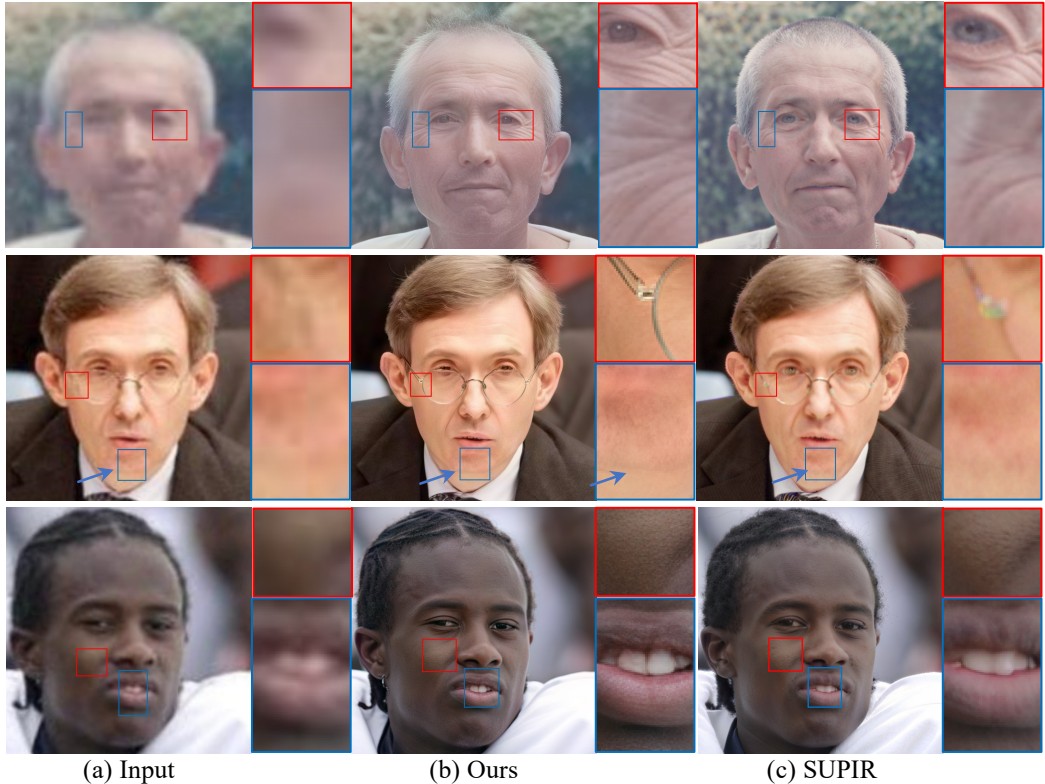

(a) Input         (b) Ours         (c) SUPIR

Figure 1: Compared with the results from a state-of-the-art (SOTA) method SUPIR (Yu et al., 2024) using StableDiffusion-XL (SDXL) (Podell et al., 2023) as prior, our approach excels in capturing and rendering intricate facial details. For instance, our result has a more distinct jawline (see blue arrow) in the 2nd row, effectively distinguishing the jaw from the neck. Zoom in for more details.

promise as an alternative to generative adversarial networks (GANs) (Song et al., 2020) in image generation. Some approaches (Yue & Loy, 2022; Wang et al., 2023c) use pretrained DDPMs to diffuse and then denoise degraded inputs. However, their practical application is hindered by the loss of original identity and detailed facial features (Miao et al., 2024), with pretrained DDPMs also facing limitations in representational capacity.

The remarkable success of large-scale pretrained text-to-image (T2I) models (Rombach et al., 2022; Saharia et al., 2022) has provided another promising prior. Many researchers explore the potential of StableDiffusion (SD) models (Stability.ai, 2024) as the powerful prior in challenging low-level vision tasks, including real-world image super-resolution (Wang et al., 2023b; Lin et al., 2024; Wu et al., 2023a; Yu et al., 2024) and BFR (Chen et al., 2024; Gao et al., 2024). Since the face details are often lost due to the degradation and down-sampling processes of VAE (Rombach et al., 2022) in SD models, BFRffusion (Chen et al., 2024) and DiffMAC (Gao et al., 2024) rely on the facial priors within SD models to recreate these details. However, being designed for general text-to-image tasks, SD models often fail to retain essential facial details, like skin texture (see Fig. 2 (b)). Therefore, these methods typically produce overly smooth images in the T2I task. Moreover, their extensive image priors can lead to the incorrect generation of non-facial features, resulting in artifacts, especially for images with ambiguous degradation. *These specific limitations – incorrect generation of non-facial features and missing facial details (red box at 3rd row of Fig. 1)– severely limit the practical deployment of these models in real-world applications.*

To tackle these problems, we propose **Authface**, a novel BFR method with face-oriented generative diffusion prior, designed to restore highly authentic face images. The **highlight** of our Authface is that it brings a paradigm shift for BFR – with a two-stage training pipeline: 1) *Face-oriented Fine-tuning on Pretrained T2I Model*, and 2) *Highly Authentic Face Restoration*. The underlying premise for Stage I is that pretrained T2I models *e.g.*, SD models, can serve as effective generative diffusion priors for restoration tasks (Sec. 3.1). They can be customized for face-centric applications via fine-tuning while retaining their generation capabilities.

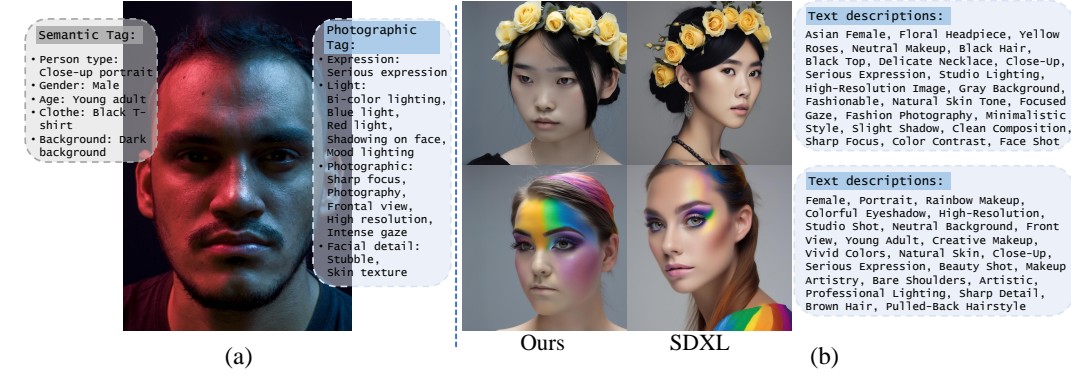

Figure 2: (a) A HQ face image with its paired tags generated through photography-guided image annotation. Specifically, we provide an additional photographic tag (blue box) beyond the semantic tags used in previous methods (gray box). (b) Qualitative comparison between StableDiffusion-XL (SDXL) (Podell et al., 2023) and our fine-tuned model, which is exclusively trained on the collected high-quality dataset, in the T2I task. Notably, SDXL tends to generate over-smooth skin even when given prompts specifying sharp details and sharp focus. Zoom in for more details.

In analyzing key factors for fine-tuning pretrained T2I models to meet human preferences for authentic facial images, we identify two key criteria as our face-oriented generative diffusion prior: 1) **Quality-first image collection**. Contrary to training T2I base models with large datasets like LAION-5B (Schuhmann et al., 2022), the quality of the dataset, rather than its size, dictates the generation quality in the fine-tuning process. 2) **Photography-guided image annotation**. Fine-tuning the pretrained T2I models for HQ facial tasks requires more than just basic annotations like human accessories, especially for HQ face images with a pronounced stylistic orientation (see Fig. 2 (a)). In line with our established criteria, we collect a curated dataset of **1.5K** HQ face images – each enriched with detailed photographic annotations – to fine-tune the pretrained T2I models for the first stage. With the curated dataset, we are able to fine-tune the T2I models following their original optimization strategies, as illustrated in Fig. 3. With fine-tuning, the pretrained T2I model is required with the detailed facial prior, which can be demonstrated with the T2I task as shown in Fig. 2 (b). To achieve the goal of highly authentic face restoration in Stage II, we leverage the ControlNet (Zhang et al., 2023) for training (Sec. 3.2). However, directly following the protocol of training ControlNet with the MSE loss tends to contribute to the loss of key facial details, such as eyes and mouths. To resolve this issue, we propose a **time-aware latent facial feature loss** to directly constrain the regions where humans are sensitive in the latent space. Our extensive experiments demonstrate the superior authentic detail generation performance on synthetic and real-world datasets.

In summary, our major contributions are three-fold: **I**) **Novel Research Direction**: Our work pioneers a new approach by enhancing the generative capabilities of pretrained T2I models for authentic face restoration, moving beyond traditional model design. **II**) **New Methodology**: Our AuthFace, a novel framework, enhances the detail handling of pretrained T2I models through a unique face-oriented restoration tuning pipeline. Our method significantly sharpens fine facial details and includes a time-aware latent facial feature loss, which effectively reduces artifacts in critical areas like the eyes and mouth. **III**) **New High-quality Dataset**: We have compiled a dataset of **1.5K** high-resolution images. We expect it can serve as a foundational and important resource to further advance the field of high-fidelity authentic face restoration.

## 2 RELATED WORKS

**Prior-based Blind Face Restoration** Blind face restoration (BFR) employs a variety of priors, classified into geometric, reference, and generative categories. Geometric priors, such as facial landmarks (Bulat & Tzimiropoulos, 2018; Kim et al., 2019; Chen et al., 2018; Ma et al., 2020), face parsing maps (Chen et al., 2021; Shen et al., 2018; Yang et al., 2020; Chen et al., 2018), facial component heatmaps (Yu et al., 2018), and 3D face shapes (Hu et al., 2020; Zhu et al., 2022; Ren et al., 2019), provide crucial structural information for restoring degraded faces. Reference-based methods use images to deliver identity information, enhancing the fidelity of the restored faces (Dogan et al., 2019; Li et al., 2020a;b; 2018). Moreover, some researchers have implemented generative facial priors, like StyleGAN (Karras et al., 2020), to refine facial details (Chen et al., 2021; Wang

et al., 2021; Chan et al., 2022; Xie et al., 2023; Wang et al., 2022a). Another approach involves using pretrained Vector-Quantize codebooks that contain detailed facial information (Wang et al., 2022b; Zhou et al., 2022; Tsai et al., 2023). Given their remarkable performance in image generation, denoising diffusion probabilistic models (Ho et al., 2020) have become increasingly popular in BFR. Notable examples, such as DifFace (Yue & Loy, 2022), DR2 (Wang et al., 2023c), and PGDiff (Yang et al., 2024), utilize denoising U-Nets pretrained on HQ face datasets to achieve face restoration at pixel level. Specifically, Zhao *et al.* (Zhao et al., 2023) attempts to improve the authentic performance via feeding network with enhanced ground-truth images. Recently, large-scale pretrained text-to-image models like StableDiffusion (SD)(Stability.ai, 2024) have been employed to address the BFR problem. DiffBIR (Lin et al., 2024) leverages SD priors for real-world image super-resolution and BFR by incorporating degraded input image information in the latent space. Specifically targeting BFR, BFRfusion (Chen et al., 2024) extracts multi-scale facial features in the latent space from low-quality face images. *However, achieving authentic BFR with pretrained T2I models in the latent space remains underexplored.*

**Fine-Tuning** Fine-tuning is widely used to align pretrained large language models (LLMs) with human preferences, improving their effectiveness (Betker et al., 2023). This technique, successful in LLMs with small, HQ datasets (Touvron et al., 2023; Zhou et al., 2024), has been adapted to text-to-image models to enhance text-image alignment (Dai et al., 2023; Li et al., 2024a;b). For example, Emu (Dai et al., 2023) improves aesthetic alignment using fine-tuned HQ image-text pairs. Playground v2.5 (Li et al., 2024a) enhances human features using a quality-controlled dataset, and CosmicMan (Li et al., 2024b) generates superior human-centric content with large, refined datasets. *However, these methods often produce overly smooth images, which may not be ideal for BFR tasks where authentic and realistic images are essential.*

## 3 METHODOLOGY

The goal of our work is to achieve authentic face restoration by minimizing unrealistic outcomes and enhancing the rendition of human-preferred features. It is structured into two distinct stages: **1)** *Face-oriented Tuning on Pre-trained T2I Model*. We integrate supervised fine-tuning (Ouyang et al., 2022) and quality-tuning (Dai et al., 2023) strategies to refine StableDiffusion-XL (SDXL), enhancing it with detailed facial features as our face-oriented generative diffusion prior (Sec. 3.1); **2)** *Highly Authentic Face Restoration*. Utilizing the face-oriented generative diffusion prior, we implement ControlNet (Zhang et al., 2023) to direct the restoration process based on the quality of input degradation (Sec.3.2). Moreover, we introduce a time-aware latent facial feature loss to improve key facial features during restoration (Sec.3.2).

### 3.1 STAGE I: FACE-ORIENTED FINE-TUNING ON PRE-TRAINED T2I MODEL

The face-oriented tuning procedure for a pre-trained T2I model consists of two main parts: 1) a quality-first dataset preparation process to obtain and filter HQ face images, and 2) photography-guided data annotation to move beyond basic labels that only convey semantic information.

**Quality-first Image Collection.** Training T2I models typically requires large datasets. However, akin to the significant performance improvements seen in large language models fine-tuned with just 1K HQ examples (Zhou et al., 2024), it has been demonstrated that enhancing the aesthetic quality of generated results can be achieved with only a few thousand extremely HQ images (Dai et al., 2023). This highlights that dataset quality is more important than size in the fine-tuning process. Inspired by this, we apply the quality-first principle to prepare our fine-tuning dataset.

Collecting HQ real-world face images is challenging due to privacy and copyright concerns, and existing datasets like FFHQ (Karras et al., 2019) suffer from issues such as JPEG degradation, blur, and Gaussian noise (Zhao et al., 2023). To overcome these challenges, we source extremely HQ face images from the professional photography website Unsplash (uns, 2023), which offers a license supporting both commercial and non-commercial use. Although the collected images are captured and post-processed by professional photographers, not all images prominently feature faces. To address this, we implement a set of data filtering strategies to create an HQ, face-centric subset. These strategies include face detection to remove images without faces or with small faces, and image quality assessment to filter out images with excessive artifacts such as pepper noise. Also, we use face landmark detection to locate eyes and mouth, enabling us to follow the alignment process used in FFHQ (Karras et al., 2019; Kazemi & Sullivan, 2014), better suited for the BFR task.

Recognizing the racial imbalance in our dataset (predominantly Caucasian and African descent), we collaborate with professional photographers to build an HQ dataset featuring individuals of Asian descent, using top-level studio settings. All facial images are manually filtered to ensure they present clear skin texture and hair details, resulting in a fine-tuning dataset of 1,500 extremely HQ images.

**Photography-guided Image Annotation.** The quality of prompts is essential for both training (Betker et al., 2023) and fine-tuning (Li et al., 2024b) pretrained T2I models. For example, CosmicMan (Li et al., 2024b) fine-tunes SDXL for human-centric content generation by breaking human parsing maps into several parts to provide detailed annotations. However, for face-oriented tuning tasks, densely annotated images are less effective. After cropping and alignment, the semantic information in facial images is limited, often capturing only overall human attributes. This differs significantly from other tasks where densely packed semantic information is prevalent. For face-oriented tuning, capturing stylistic information beyond basic semantics is crucial. In portrait photography, this includes expressions, skin texture, makeup, and lighting, essential for authentic face restoration.

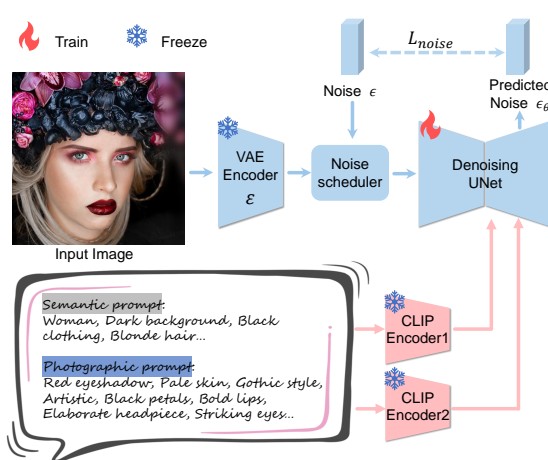

Figure 3: The framework of face-oriented tuning.

Therefore, we apply photography-guided data annotation to generate prompts for our fine-tuning dataset, especially given that our dataset consists of HQ portraits by professional photographers with strong stylistic tendencies. We follow previous methods (Betker et al., 2023; Li et al., 2024b) to realize automatic captioning tasks with Vision-Language Models (VLMs). Specifically, we leverage the pretrained LLaVA-1.6 (Liu et al., 2024) as the automatic caption to generate a tag-style prompt to avoid redundant prepositions and adverbs (Hertz et al., 2022). Fig. 2 (a) illustrates some examples of photography-guided data annotation. Based on the dataset, we can fine-tune the pre-trained T2I model, SDXL, as shown in Fig. 3. Different from the training of SDXL, we fix the resolution of training images instead of multi-aspect training.

## 3.2 STAGE II: HIGHLY AUTHENTIC FACE RESTORATION

Fig. 4 illustrates the structure of stage II. Given the fine-tuned SDXL model as our face-oriented generative diffusion prior, we need an adaptor that can control the fine-tuned SDXL to generate high-quality facial images based on its degraded input. With the successful application of ControlNet (Zhang et al., 2023) in real-world image super-resolution (Wu et al., 2023a; Yu et al., 2024), we apply it as the controller for the BFR task.

The training of stage II is as follows. The latent representation of an HQ facial image is obtained by the encoder of a pretrained VAE, denoted as $\mathbf{z}_0$. The diffusion process progressively introduces noise to $\mathbf{z}_0$, resulting in a noisy latent $\mathbf{z}_t$, where $t$ represents the randomly sampled diffusion step. The restoration is conditioned on the additional input $\mathbf{c}$, which is the degraded face image, guiding the generation process. For each diffusion step $t$, the noisy latent $\mathbf{z}_t$ is processed together with the control condition $\mathbf{c}$, and null prompts [""]. We train the ControlNet by minimizing the $L_2$ loss between the predicted noise $\epsilon_\theta$ and the added noise $\epsilon$ ($\epsilon \sim \mathcal{N}(0, I)$). The optimization objective is:

$$\mathcal{L}_{noise} = \mathbb{E}_{\mathbf{z}_0, t, \mathbf{c}, \epsilon \sim \mathcal{N}(0,I)} \left[ \| \epsilon - \epsilon_\theta(\mathbf{z}_t, t, [""], \mathbf{c}) \|_2^2 \right]. \quad (1)$$

Specifically, we freeze the parameters of our fine-tuned SDXL model to preserve the enhanced facial priors and its original natural image priors. We initialize ControlNet with the encoder from our fine-tuned SDXL model while solely training ControlNet.

**Time-aware Latent Facial Feature Loss.** Reducing incorrect generation is crucial for authentic face restoration, as humans are sensitive to key facial features like eyes and mouths. However, the MSE loss (Eq. 1) used to train the ControlNet only provides a holistic constraint, where both the background and face of the degraded image equally influence optimization. Thanks to the spatially

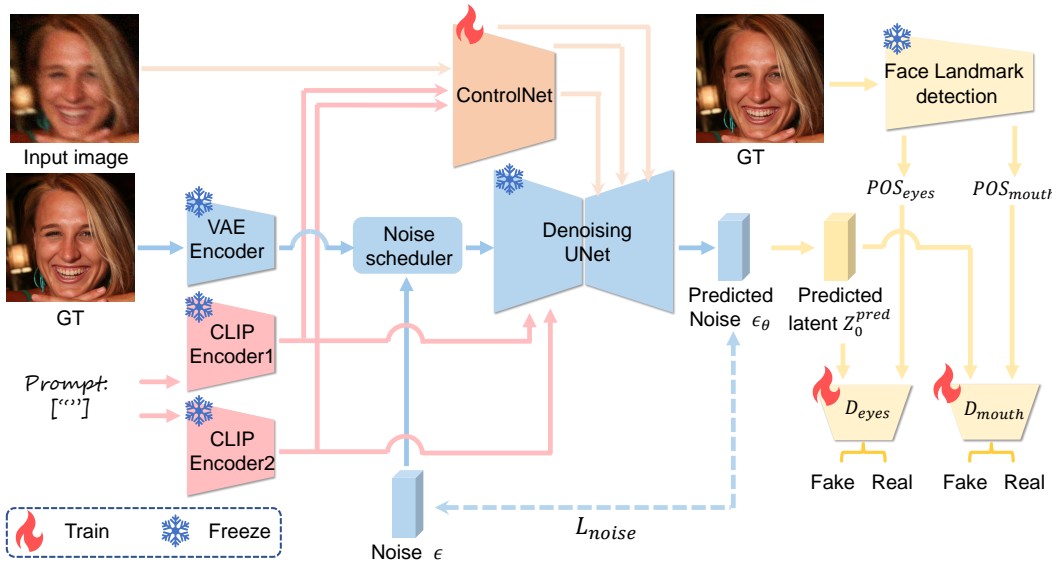

Figure 4: An overview of Stage II. Denoising UNet, carried over from Stage I, maintains its facial priors by freezing its parameters, while ControlNet acts as an adapter for handling degraded inputs.

invariant features of the conditioning embedding module in ControlNet, the latent space retains spatial dimensions (Avrahami et al., 2023). This allows for pixel-level constraints in the latent space.

To enhance key facial features, we propose a time-aware latent facial feature loss that provides additional constraints on the eyes and mouth. Inspired by GFP-GAN (Wang et al., 2021), we train separate facial feature discriminators to ensure these regions in the restored results match natural distributions. Unlike GFP-GAN, our method incorporates the diffusion and denoising process of DDPMs, *considering time as a variable*. Previous studies(Wang et al., 2023b; Avrahami et al., 2023; Choi et al., 2022) show that during denoising, the generated results evolve from rough shapes to high-resolution images. Therefore, using shared model weights for various time steps is not ideal. Given the predicted noise $\epsilon_\theta$, sampled diffusion step $t$, and noisy latent $\mathbf{z}_t$, we can estimate the predicted latent $\mathbf{z}_0^{\text{pred}}$ according to the closed form formulation in DDIM (Song et al., 2020) as:

$$\mathbf{z}_0^{\text{pred}} = \frac{\mathbf{z}_t - \sqrt{\beta_{\text{prod}_t}} \cdot \epsilon_\theta}{\sqrt{\alpha_{\text{prod}_t}}}, \alpha_{\text{prod}_t} = \prod_{i=1}^{t} \alpha_i \quad , \quad \beta_{\text{prod}_t} = \prod_{i=1}^{t} \beta_i \tag{2}$$

where $\alpha_i$ is the noise decay factor at each diffusion step and $\beta_i$ is the noise variance schedule. We then locate these two regions with pertrained face landmark detection network (Zheng et al., 2021) from the ground-truth (GT) image, and transform these pixel level's location to the latent space ones $(POS_{eyes}, POS_{mouth})$ by downsampling with a factor of eight. With the latent space position of eyes and the mouth, we crop these regions from the predicted latent $\mathbf{z}_0^{\text{pred}}$ and the latent of HQ image $z_0$ respectively to obtain the facial feature patches, $P_{eye} = \{p_{eye}, p_{eye}^{pred}\}$ and $P_{mouth} = \{p_{mouth}, p_{mouth}^{pred}\}$. Inspired by the logit-normal sampling in StableDiffusion 3 (Esser et al., 2024) and the finding in Fig. 5, our time-aware latent facial feature loss focus on the intermediate steps when the major shape of eyes and mouth arise via assigning higher weight, as follow:

$$\text{Weight}P = \frac{1}{s\sqrt{2\pi}} \frac{1}{t(1-t)} \exp\left(-\frac{(\text{logit}(t) - m)^2}{2s^2}\right), \tag{3}$$

where $\text{logit}(t) = log\frac{1}{t(1-t)}$, $m$ and $s$ are the location parameter and scale parameter, respectively. The time-aware latent facial feature loss is defined as follows:

$$\mathcal{L}_{facial} = \sum_{P \in P_{\text{eye}}, P_{\text{mouth}}} \text{Weight}P\Big(\lambda_d \mathbb{E}_{p^{\text{pred}}} \left[\log(1 - \mathbb{D}_P(p^{\text{pred}}))\right]$$

$$+ \lambda_s \left|\text{Gram}(\psi(p^{\text{pred}})) - \text{Gram}(\psi(p))\right|\Big), \tag{4}$$

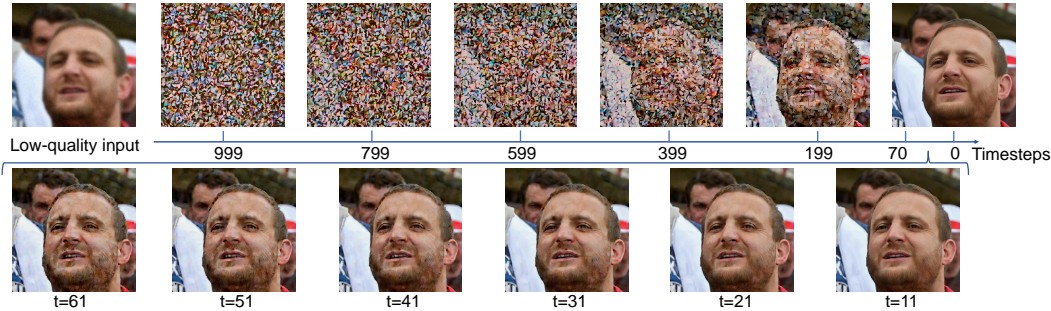

Figure 5: Visualization of the diffusion process at different steps. In the early steps (t = 999 - 599), the main content of the images is predominantly noise, with key facial features obscured. In the later steps (t = 61 - 0), the shapes of key facial features become fixed, with minimal changes.

where $\mathbb{D}_P$ refers to the discriminators for different facial regions, specifically $\mathbb{D}_{eyes}$ and $\mathbb{D}_{mouth}$. $\psi$ represents multi-scale features of the regional facial feature discriminator. Gram() operation refers to calculating the Gram matrix static (Gatys et al., 2016) $\lambda_d$ and $\lambda_s$ are the weights of the discriminative loss and the style loss, respectively. The total loss function of AuthFace is defined as $\mathcal{L}_{\text{total}} = \mathcal{L}_{noise} + \mathcal{L}_{facial}$.

## 4 EXPERIMENTS

### 4.1 EXPERIMENTAL SETTINGS

**Implementation Details:** Our base model is initialized from StableDiffusion-XL (SDXL) (Podell et al., 2023), and we fine-tune the entire U-Net from this base model. We employ AdamW (Loshchilov & Hutter, 2017) optimizer with the learning rate of $5e - 7$ during the fintuing process, where the batch size and the training iteration are set to 96 and 50k, respectively. We apply the same optimizer with the learning rate of $2e - 5$ with the batch size 48 for training ControlNet. All experiments are conducted on four NVIDIA L40s GPUs in the resolution of 1024 $\times$ 1024 for fintuning model and 512 $\times$ 512 for training ControlNet.

**Training and Test Dataset:** The training dataset for the fine-tuning process of our face-oriented model comprises **1.5K** high-quality face images, each enriched with detailed photographic annotations. For training our AuthFace network, we resize the FFHQ dataset (Karras et al., 2019) from a resolution of 1024×1024 to 512×512. To form training pairs, we follow the settings, including degradation types and degrees, as outlined in previous methods (Wang et al., 2021; Chen et al., 2024). Following (Zhou et al., 2022; Zhao et al., 2023; Yang et al., 2024), we evaluate our method on a synthetic dataset, CelebA-Test (Liu et al., 2015), and three real-world datasets: LFW-Test (Wang et al., 2021), WebPhoto-Test (Wang et al., 2021), and WIDER-Test (Zhou et al., 2022).

**Metrics:** To evaluate our method's performance on the Celeb-A dataset with ground truth, we use PSNR (Hore & Ziou, 2010), SSIM (Wang et al., 2004), and LPIPS (Zhang et al., 2018). Besides, we follow SUPIR (Yu et al., 2024) introducing non-reference image quality assessment metrics, MUSIQ (Ke et al., 2021), ManIQA (Yang et al., 2022) ,ClipIQA (Wang et al., 2023a), and FID (Heusel et al., 2017).

### 4.2 COMPARISON AND EVALUATION

We compare our method with SOTA BFR methods in three different categories: **(I)** GAN-based methods, including GFP-GAN (Wang et al., 2021) and PSFR-GAN (Chen et al., 2021); **(II)** Codebook-based method, including CodeFormer (Zhou et al., 2022); **(III)** Diffusion-based methods, including DR2 (Wang et al., 2023c) and BFRffusion (Chen et al., 2023); Notably, we compare our method with SOTA IR method, SUPIR (Yu et al., 2024), which is also based on SDXL (Podell et al., 2023). All methods are tested with official codes.

**Comparison on Synthetic Dataset:** Quantitative results in Tab. 1 showcase our method's superior performance on the CelebA-Test dataset, outperforming baselines in all non-reference image quality assessment metrics except FID. Notably, we achieve SOTA performance in terms of the LPIPS score.

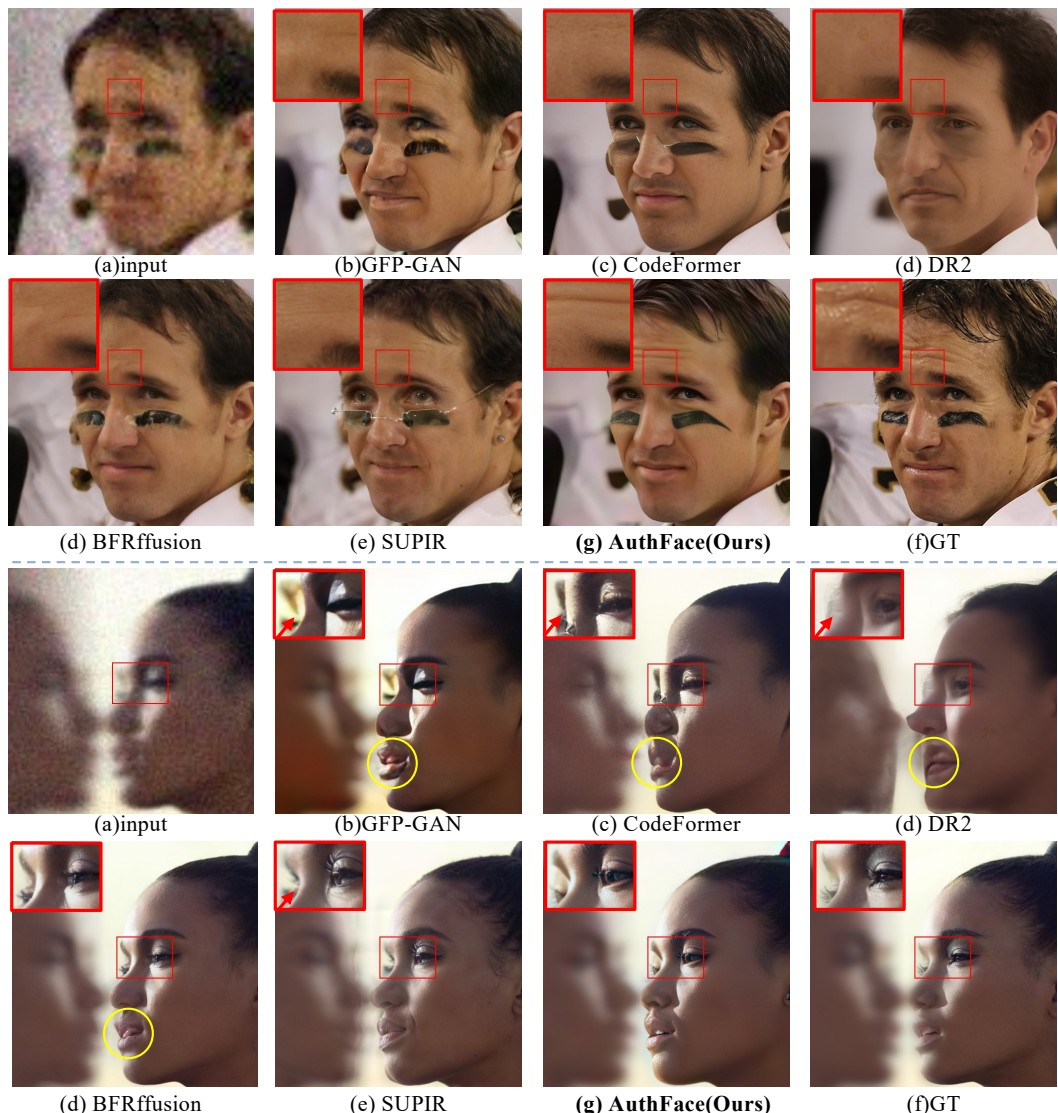

Figure 6: Qualitative results on CelebA-Test dataset. Red box areas in 1st row highlight the detailed skin texture and eyebrows achieved by our method. **Zoom in for details.**

This marks a significant validation of our approach for the BFR task. Qualitatively, as depicted in Fig. 6, our AuthFace achieves authentic face restoration. Specifically, all methods except ours fail to recover the face paint in the first example, and our method results in the best hair detail. The second example involves recovering the side face, which is one of the most challenging cases in BFR (Zhou et al., 2022). GFP-GAN, CodeFormer, and BFRfusion fail to restore authentic mouth details (yellow circle), while the results from GFP-GAN, CodeFormer, DR2, and SUPIR lose the right eye (red box). Only our method produces realistic results in these key regions, with best details in the eyebrow and skin texture.

**Comparison on Real-world Dataset:** The robustness of our method is demonstrated by its SOTA performance in all metrics and real-world datasets, except for the FID score in the LFW-Test and WebPhoto-Test datasets, as shown in Tab. 1. Notably, the MANIQA score in the LFW-Test dataset exceeds the baselines by **0.09**. In the LFW-Test dataset, GFP-GAN, CodeFormer, and DR2 fail to reconstruct realistic results in the eye regions due to incorrect generation at the edges of glasses (see the red box in the first row of Fig. 7). In the second row of Fig. 7, our method outperforms others by accurately reconstructing both the upper and lower teeth without the artifacts around the hands, highlighted in a yellow circle. In the WebPhoto-Test dataset, our approach not only precisely

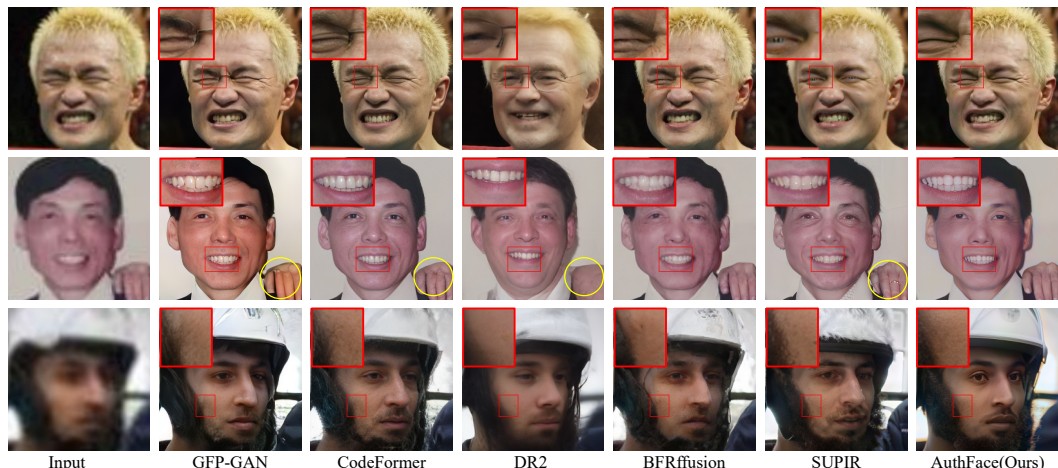

Input     GFP-GAN     CodeFormer     DR2     BFRffusion     SUPIR     AuthFace(Ours)

Figure 7: Qualitative results on real-world datasets. Results in 1st row are from LFW-Test dataset (Wang et al., 2021). Results in 2nd row come from WebPhoto-Test dataset (Wang et al., 2021). Results in 3rd row are from WIDER-Test dataset (Zhou et al., 2022) including a zoomed-in view of the skin highlighted in red box areas. **Zoom in for details.**

Table 1: Quantitative results for blind face restoration on both synthetic and real-world datasets. The highest result is highlighted in **red** while the second highest result is highlighted in blue.

| Datasets | Metrics | GFPGAN | PSFRGAN | CodeFormer | DR2 | BFRffusion | SUPIR | Ours |
|---|---|---|---|---|---|---|---|---|
| CelebA | PSNR↑ | 24.65 | 24.68 | 25.15 | 21.43 | **26.19** | 25.00 | 25.57 |
| | SSIM↑ | 0.6669 | 0.6322 | 0.6647 | 0.5943 | **0.6829** | 0.6487 | 0.6768 |
| | LPIPS↓ | 0.2308 | 0.2943 | 0.2269 | 0.3443 | 0.2272 | 0.2716 | **0.2143** |
| | MANIQA↑ | 0.5633 | 0.5103 | 0.5546 | 0.5397 | 0.5964 | 0.5233 | **0.6624** |
| | MUSIQ↑ | 73.91 | 73.32 | 75.56 | 70.42 | 71.90 | 72.92 | **75.76** |
| | FID↓ | 42.62 | 47.59 | 52.43 | 56.59 | 40.74 | **35.01** | 50.93 |
| | CLIPIQA↑ | 0.6790 | 0.6310 | 0.6716 | 0.5770 | 0.6863 | 0.6103 | **0.7065** |
| LFW | MANIQA↑ | 0.5514 | 0.5176 | 0.5415 | 0.5326 | 0.5528 | 0.4768 | **0.6431** |
| | MUSIQ↑ | 73.58 | 73.60 | 75.49 | 71.04 | 69.86 | 69.90 | **75.87** |
| | FID↓ | 49.96 | 51.89 | 52.36 | 47.14 | 49.92 | **41.98** | 45.29 |
| | CLIPIQA↑ | 0.6994 | 0.6471 | 0.6893 | 0.6069 | 0.6969 | 0.5931 | **0.7350** |
| WebPhoto | MANIQA↑ | 0.5351 | 0.4793 | 0.5241 | 0.4843 | 0.4721 | 0.4394 | **0.5860** |
| | MUSIQ↑ | 72.13 | 71.67 | 74.01 | 67.19 | 61.78 | 65.67 | **74.11** |
| | FID↓ | 87.35 | 88.45 | 83.19 | 107.86 | 84.29 | **73.44** | 90.04 |
| | CLIPIQA↑ | 0.6888 | 0.6366 | 0.6922 | 0.5690 | 0.6308 | 0.5767 | **0.6964** |
| WIDER | MANIQA↑ | 0.5289 | 0.4925 | 0.5119 | 0.4989 | 0.4923 | 0.4522 | **0.5941** |
| | MUSIQ↑ | 72.80 | 71.50 | 73.40 | 67.18 | 61.87 | 67.19 | **74.59** |
| | FID↓ | 39.49 | 49.84 | 38.78 | 45.27 | 55.22 | 42.61 | **36.10** |
| | CLIPIQA↑ | 0.7101 | 0.6482 | 0.6990 | 0.5943 | 0.6789 | 0.6093 | **0.7306** |

reconstructs details such as helmets and goatees but also delivers the best skin texture, as showcased in the red box areas. *More visualization results are in the appendix and the supplmat.*

## 4.3 ABLATION STUDY

**Effectiveness of Face-oriented Fine-Tuning:** We conducted an ablation study to evaluate the effectiveness of face-oriented tuning on CelebA-Test and WebPhoto-Test, as shown in Tab. 2 (a) and (b). In experiment (a), the original SDXL is used as the base model, and ControlNet is initialized with it. In experiment (b), the fine-tuned SDXL is used as the base model, and ControlNet is initialized with this fine-tuned version. Except for the MANIQA score on the CelebA-Test dataset, experiment (b) consistently outperforms experiment (a), highlighting the necessity of face-oriented tuning for the generative diffusion prior. Notably, in the WebPhoto-Test dataset, experiment (b) excels across all metrics, including CLIPIQA (0.6833 vs. 0.6276), MUSIQ (72.35 vs. 67.01), and MANIQA (0.5810 vs. 0.5252). Experiment (b) enhances facial details such as eyebrows and skin texture (red box, 1st row in Fig. 8) and eyelashes (red box, 2nd row). Additionally, it reduces errors in key facial features, resulting in clearer eyes (red box, 1st row) and better restoration of teeth (blue box, 2nd row).

Table 2: Ablation studies of variant generative diffusion prior and time-aware latent facial feature loss. The highest result is highlighted in **red** while the second highest result is highlighted in blue.

| Dataset | Exp. | Diffusion Prior | | $\mathcal{L}_{facial}$ | | Metrics | | | |
|---|---|---|---|---|---|---|---|---|---|
| | | SDXL | Ours | Const. | Time-aware | PSNR↑ | MANIQA↑ | MUSIQ↑ | CLIPIQA↑ |
| CelebA | (a) | ✓ | | | | 24.39 | 0.5781 | 69.25 | 0.6465 |
| | (b) | | ✓ | | | **25.59** | 0.5057 | 74.42 | **0.7088** |
| | (c) | | ✓ | ✓ | | 23.95 | 0.6449 | 73.66 | 0.6821 |
| | (d) | | ✓ | | ✓ | 25.57 | **0.6624** | **75.76** | 0.7065 |
| WebPhoto | (a) | ✓ | | | | - | 0.5252 | 67.01 | 0.6276 |
| | (b) | | ✓ | | | - | 0.5810 | 72.35 | 0.6833 |
| | (c) | | ✓ | ✓ | | - | 0.5767 | 68.52 | 0.6861 |
| | (d) | | ✓ | | ✓ | - | **0.5860** | **74.11** | **0.6964** |

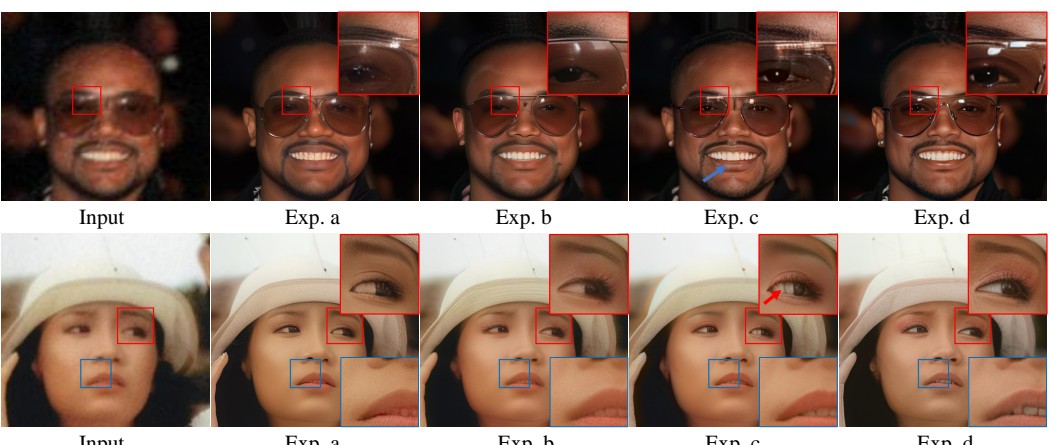

Figure 8: Visualization of ablation results. 1$^{st}$ row and 2$^{nd}$ row are the examples from CelebA-Test and WebPhoto-Test datasets, respectively. Please zoom in for more details.

**Effectiveness of Time-aware Latent Facial Feature Loss:** To evaluate the effectiveness of the time-aware latent facial feature loss, we conducted experiments as shown in Tab. 2 (b), (c), and (d). Using constant weights for various time steps (experiment (c)) negatively impacts optimization, resulting in performance drops across most metrics, except for the MANIQA score on the CelebA-Test dataset compared to experiment (b). By focusing on steps when the major shapes of eyes and mouth emerge and assigning higher weights during these steps, our time-aware loss achieves the best performance on both synthetic and real-world datasets, except for the PSNR and CLIPIQA scores on the CelebA dataset. As shown in Fig. 8, using latent space facial feature loss in experiments (c) and (d) improves the restoration of eyes and mouth (see the red box in the first row and the blue box in the second row). Notably, the time-aware strategy in experiment (d) not only reduces artifacts (as indicated by the blue and red arrows in Fig. 8) but also enhances details (e.g., the sharp edge of the glasses in the first row and the delicate skin texture and eyebrows in the 2$^{nd}$ row).

## 5 CONCLUSION

This paper presented a new approach for achieving authentic face restoration by avoiding incorrect generations and enhancing facial details. Specifically, we proposed a face-oriented restoration-tuning paradigm to fine-tune the pretrained T2I model with high-quality face images, enabling the pretrained T2I model, SDXL, to develop a prior for facial details. Utilizing this face-oriented generative diffusion prior, we introduced AuthFace for the blind face restoration task, achieving authentic face restoration. Additionally, we introduced the time-aware latent facial feature loss to further improve the robustness of restoration in key facial features. Experimental results demonstrate the superiority and effectiveness of our method.

**Limitation and Future Work:** The process of collecting high-quality images requires significant human resources to filter out low-quality images. We plan to develop an aesthetic-oriented image quality assessment network to reduce labor costs.

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

## A  DETAILS OF FACE-ORIENTED TUNING DATASETS

Existing datasets like FFHQ (Karras et al., 2019) suffer from issues such as JPEG degradation, blur, and Gaussian noise (Zhao et al., 2023). We have compiled a collection of 1,500 high-quality images, exceeding resolutions of 8K, captured by professional photographers. These improvements address the quality limitations of traditional datasets, with examples showcased in Fig. 9 and Fig. 10.

We detail our dataset collection and annotation process in Fig. 11, which depicts the entire pipeline, including data collection, reviewing, and tagging. Apart from sourcing from Unsplash, we collaborate with professional photographers to collect studio-based images of Asian descent. These professionals also retouch each image to enhance skin texture while removing blemishes. All images undergo manual screening to ensure they are neither over-smoothed nor contain pepper noise, preserving detailed facial features.

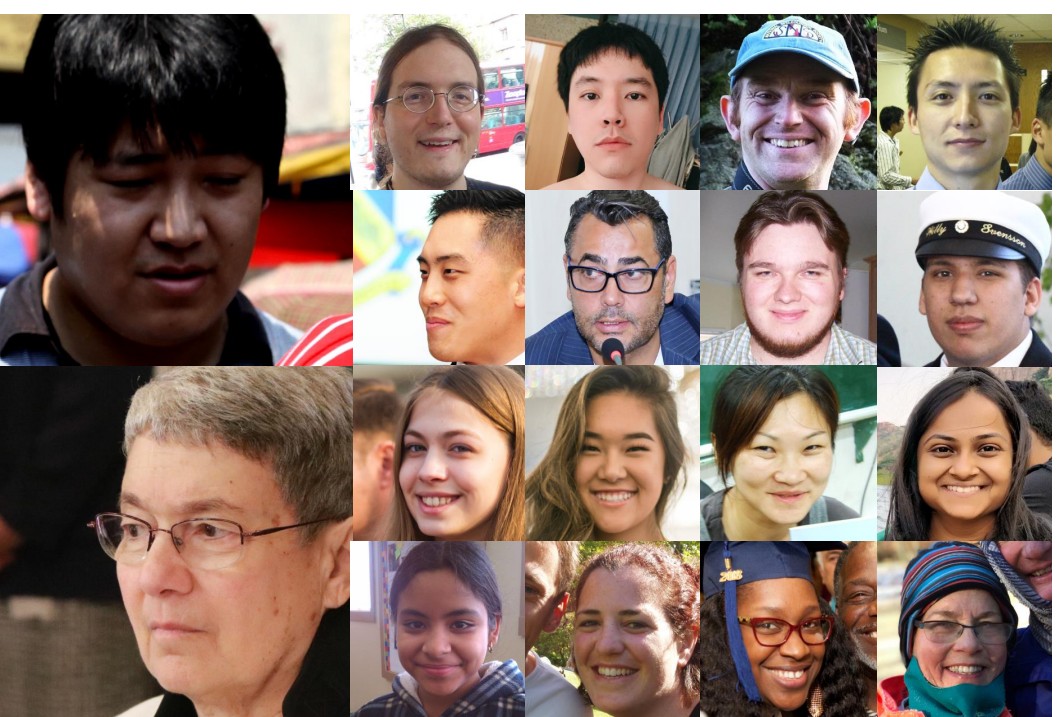

Figure 9: Example of FFHQ datasets with noticeable blur and noise. Zoom in for more details.

## B  MORE ABLATION STUDIES

**Effectiveness of Face-oriented Fine-Tuning:** To demonstrate the effectiveness of face-oriented fine-tuning, we conduct additional experiments on the task of text-to-image and provide more qualitative results on LFW-Test for the task of blind face restoration.

First, to underscore the value of our photography-guided annotation, we conducted ablation studies under three settings: (a) using the pretrained SDXL; (b) fine-tuning the SDXL with only semantic tags; and (c) fine-tuning with both semantic and our proposed photography-guided tags. We evaluated authenticity with FID and Human Preference Score v2 (Hpsv2) (Wu et al., 2023b) and through a user study with 20 participants who assessed images from 10 prompts. According to Tab. 3, setting (c) not only scored the highest on FID and Hpsv2 but also received the best average ranking, indicating that users consistently preferred images produced by models fine-tuned with both semantic and our proposed photography-guided tags. Besides, Fig. 12 demonstrates that our face-oriented fine-tuning successfully equips SDXL with dedicated facial details.

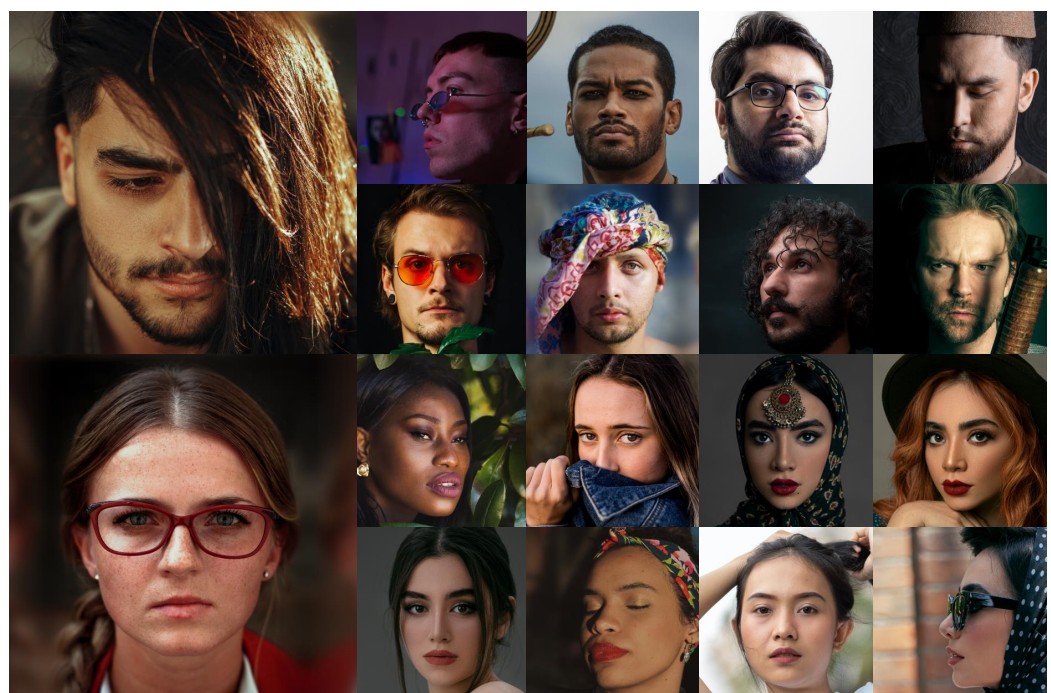

Figure 10: Example of our HQ datasets with details of skin texture. Zoom in for more details.

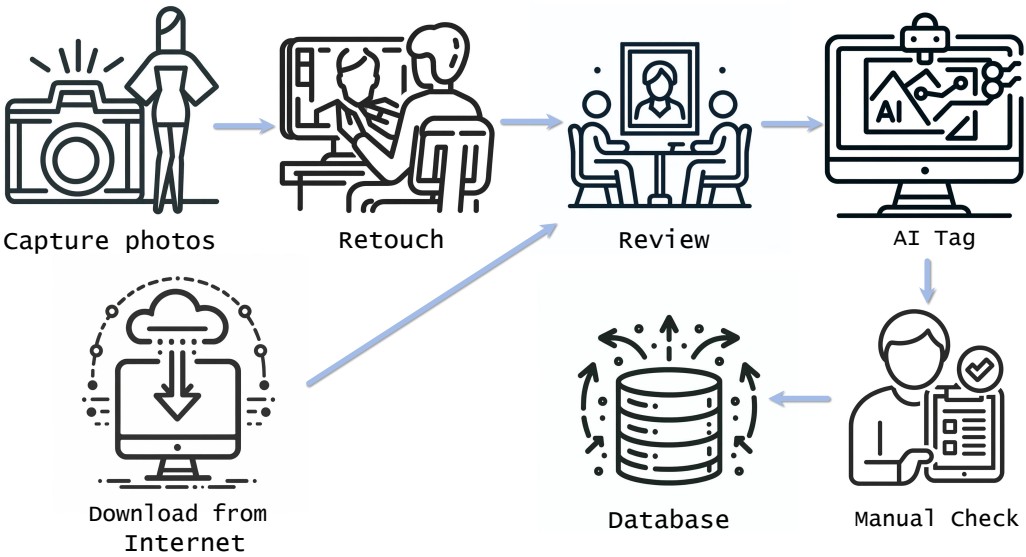

Figure 11: Illustration of collecting high-quality datasets for face-oriented fine-tuning.

To further evaluate the effectiveness of face-oriented fine-tuning We provide more qualitative results on LFW-Test as shown in Fig. 13. In experiment (a), the original SDXL is used as the base model, and ControlNet is initialized with it. In experiment (b), the fine-tuned SDXL is used as the base model, and ControlNet is initialized with this fine-tuned version. Notably, the results of Exp. b has the best visual experience enjoying authentic facial details, such as the dedicated skin texture and hair, which demonstrates the importance of face-oriented fine-tuning.

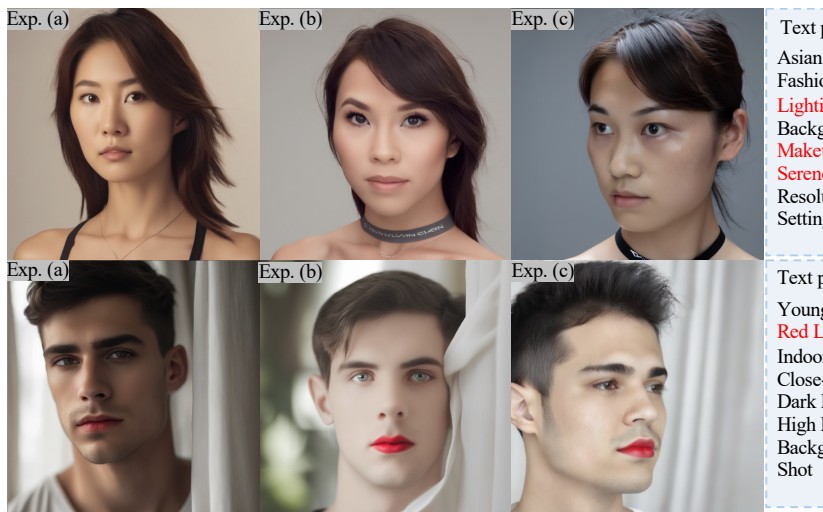

Figure 12: More qualitative comparison of the text-to-image task between (a) using the pretrained SDXL; (b) fine-tuning the SDXL with only semantic tags; and (c) fine-tuning with both semantic and our proposed photography-guided tags.

Table 3: Ablation studies of variant tags used in fine-tuning. The highest result is highlighted in The highest result is highlighted in red, while the second highest result is are blue for clarity.

| Exp | FID ↓ | Hpsv2 (Wu et al., 2023b) ↑ | User study rank ↓ |
|-----|-------|---------------------------|-------------------|
| (a) | 95.13 | 0.2637 | 2.53 |
| (b) | 62.90 | 0.2712 | 2.17 |
| (c) | **51.09** | **0.2903** | **1.33** |

**Different hyper meter of time-aware latent facial feature loss:** Through our face detection experiments, we identified that the timestep where the average confidence score reaches 0.5 is 0.32 (normalized) in the FFHQ dataset. We designated this timestep as the most critical, assigning it the highest weight. Therefore, we set the m as -0.5 and s as 1.0, where Eq. 3 in the main paper peaks at t=0.37. We conduct an ablation study on these hyperparameters m and s as detailed in Tab. 4 and we also provide Fig. 14 showing the weight distributions of different hypermeter. This study confirms that our chosen settings yield the best outcomes, thus validating the robustness of our experimental approach.

Table 4: Ablation studies of variant generative diffusion prior and time-aware latent facial feature loss. The highest result is highlighted in red while the second highest result is highlighted in blue.

| Dataset | Exp. | Location parameter m | Metrics | | | |
|---------|------|---------------------|---------|---------|--------|----------|
| | | | PSNR↑ | MANIQA↑ | MUSIQ↑ | CLIPIQA↑ |
| CelebA | (a) | m = -0.5 | **25.57** | **0.6624** | **75.76** | **0.7065** |
| | (b) | m = 0.0 | 25.40 | 0.6399 | 74.92 | 0.6786 |
| | (c) | m = 0.5 | 25.37 | 0.6462 | 74.67 | 0.6882 |
| | (d) | s = 0.5 | 25.42 | 0.6440 | 74.72 | 0.6794 |
| WebPhoto | (a) | m = -0.5 | - | 0.5860 | 74.11 | 0.6964 |
| | (b) | m = 0.0 | - | 0.5760 | 73.51 | 0.6657 |
| | (c) | m = 0.5 | - | 0.5829 | 73.40 | 0.6755 |
| | (d) | s = 0.5 | - | 0.5756 | 73.06 | 0.6686 |

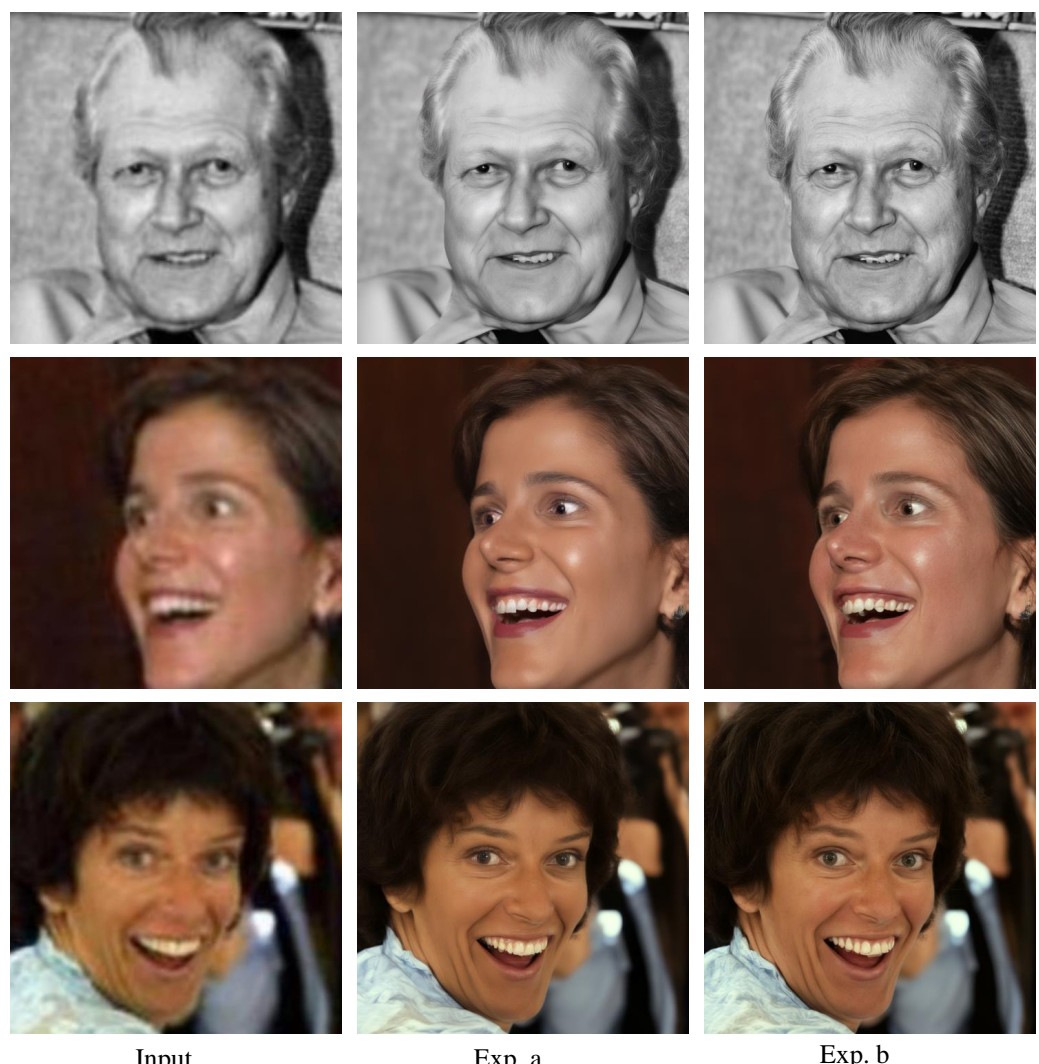

Input        Exp. a        Exp. b

Figure 13: Visualization of ablation results on LFW-Test dataset. Zoom in for more details.

## C  USER STUDY

We implemented an AB-test with 20 participants using 20 facial images from our datasets to gauge human perception of our method compared to six baselines. Participants were shown two images, labeled A and B, and asked to choose from three responses: "A is better," "B is better," or "Both are equally good," with image positions randomized. As detailed in Tab. 5, our method was preferred, indicating it produces results that are both more authentic and visually appealing.

Table 5: Results of user study. "Ours" is the percentage that our result is preferred, "Others" is the percentage that some other method is preferred, "Same" is the percentage that the users have no preference.

| Methods | Others | Same | Ours |
|---|---|---|---|
| GFPGAN | 26.75% | 2.5% | 70.75% |
| PSFRGAN | 7.5% | 0.75% | 91.75% |
| CodeFormer | 23.5% | 1.25% | 75.25% |
| DR2 | 10.5% | 0.5% | 89% |
| BFRffusion | 25.25% | 2% | 72.75% |
| SUPIR | 22.5% | 1% | 76.5% |

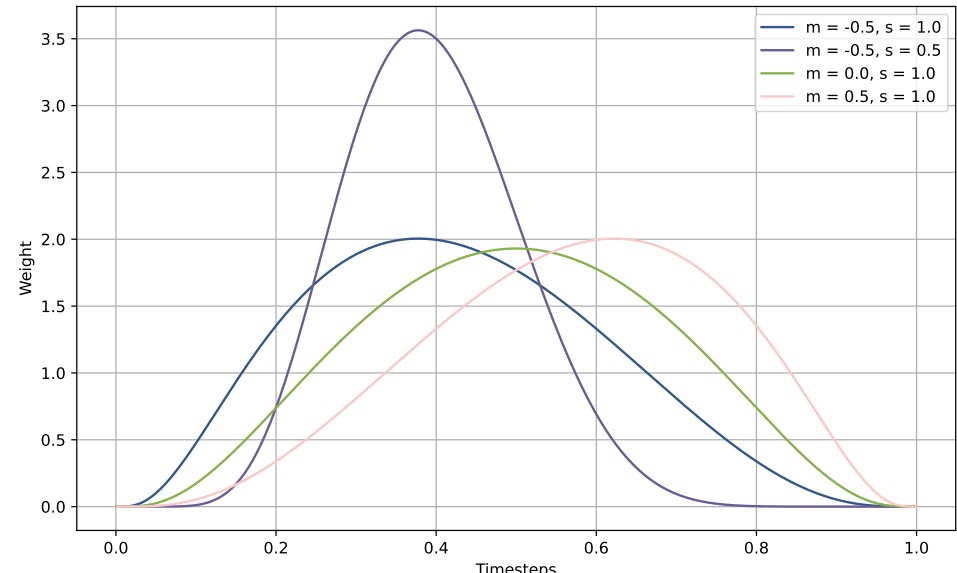

Figure 14: The weight distributions of different hyper meters of time-aware latent facial feature loss.

## D  RUNNING TIME

We evaluated our method's runtime on a single NVIDIA L40s GPU, as detailed in Tab. 6. Notably, although both our method and SUPIR utilize SDXL, SUPIR demands significantly more time due to its initial enhancement phase and the use of LLaVA for text prompts. This supports our decision to omit details prompts in stage 2.

Table 6: Running time of different networks. Please note that all methods are evaluated in $512\times 512$ input images, while DR2 reconstructs high-quality face images at $256\times 256$ and upscale to $512\times512$ with an enchantment module according to its official setting.

| Method | PSFRGAN | GFP-GAN | CodeFormer | DR2 | BFRffusion | SUPIR | Ours |
|---|---|---|---|---|---|---|---|
| Time (s) | 0.06 | 0.17 | 0.01 | 0.49 | 2.89 | 10.36 | 5.25 |

## E  MORE VISUALIZATION RESULTS

In this section, we provide more visual comparisons with state-of-the-art methods in CelebA-Test, LFT-Test, WebPhoto-Test, and WIDER-Test datasets as shown in Fig. 15 and Fig. 16.

## F  ADDITIONAL RESULTS OF OTHER RESTORATION METHODS

In this section, we provide more visual comparisons with DiffBIR (Lin et al., 2023) and StableSR (Wang et al., 2024) in CelebA-Test, LFT-Test, WebPhoto-Test, and WIDER-Test datasets as shown in Fig. 17 and Fig. 18. Our method surpasses DiffBIR and StableSR in visual quality by providing more detailed skin textures and reducing incorrect generation in key facial features. For example, the red-boxed areas in Fig. 17 and 18 showcase that DiffBIR tends to generate overly smooth skin textures. While StableSR performs better in facial detail than DiffBIR, it suffers from incorrect generation due to limitations of CodeFormer.

## G  UNIVERSAL EXPERIMENT

In this section, we conduct a comprehensive experiment to evaluate the effectiveness of our face-oriented fine-tuning by applying our face-oriented generative diffusion prior to existing methods.

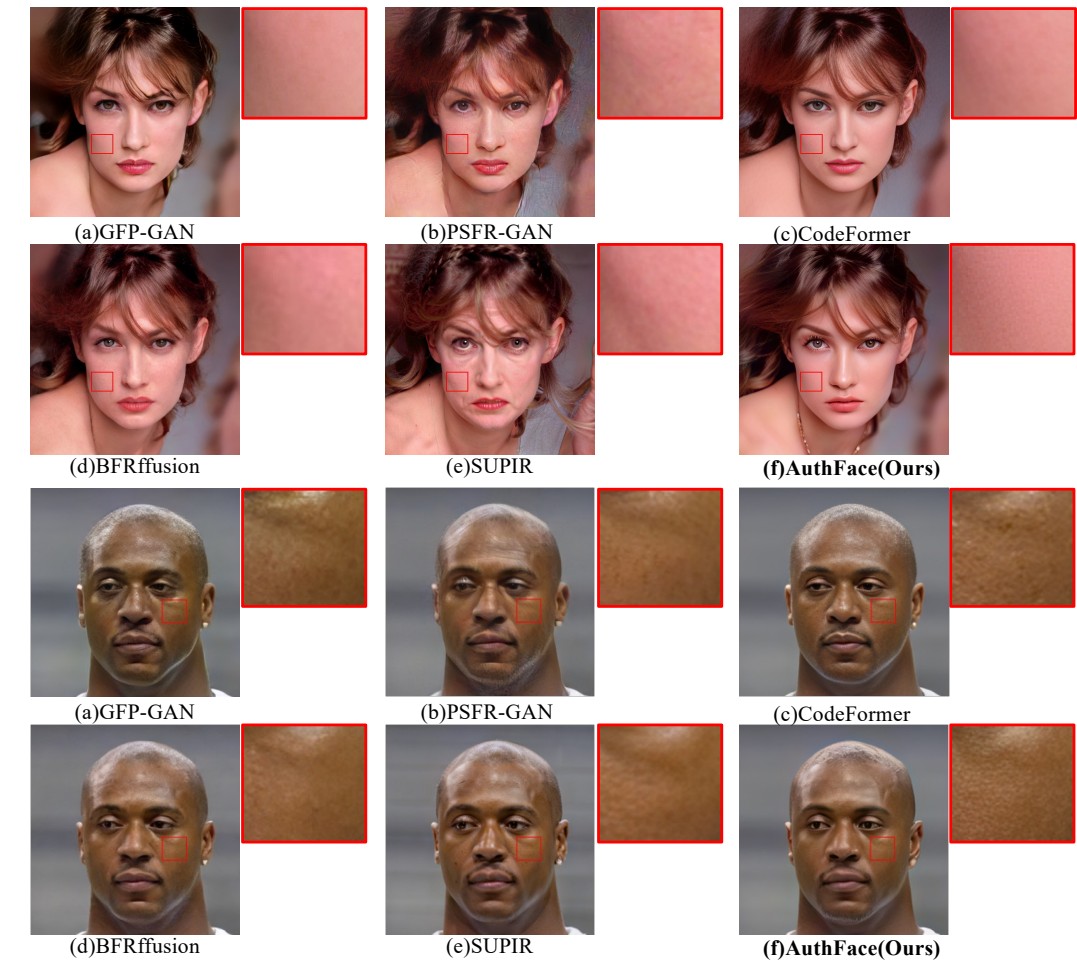

Figure 15: Visualization results in CelebA-Test and LFW-Test dataset including a zoomed-in view of the skin highlighted in red box areas. Zoom in for more details.

Since SUPIR—the only SDXL-based method—provides only its testing code and datasets, we replace the pretrained SDXL model in SUPIR with our face-oriented generative diffusion prior while keeping all other parameters from SUPIR's original model. We provide visual comparisons between the original version of SUPIR (SUPIR-original) and the version using our prior (SUPIR-auth), as shown in Fig. 19.

With our face-oriented diffusion prior, SUPIR(auth) demonstrates noticeable visual improvements over SUPIR(original) in terms of facial details, especially the skin texture, as shown in the first and third rows of Fig. 19. Furthermore, SUPIR(auth) significantly improves the restoration of key facial features, such as the eyes and mouth, thanks to the enhanced facial prior, as illustrated in the second row of Fig. 19. These findings align with our motivation of achieving authentic face restoration by providing a face-oriented prior to improve facial details and avoid incorrect generations.

However, since we only provide our fine-tuned models and all other parameters are inherited from SUPIR's original model, SUPIR's wrinkle bias still exists. Consequently, SUPIR(auth) amplifies these wrinkles to some degree, as shown in the third row of Fig. 19.

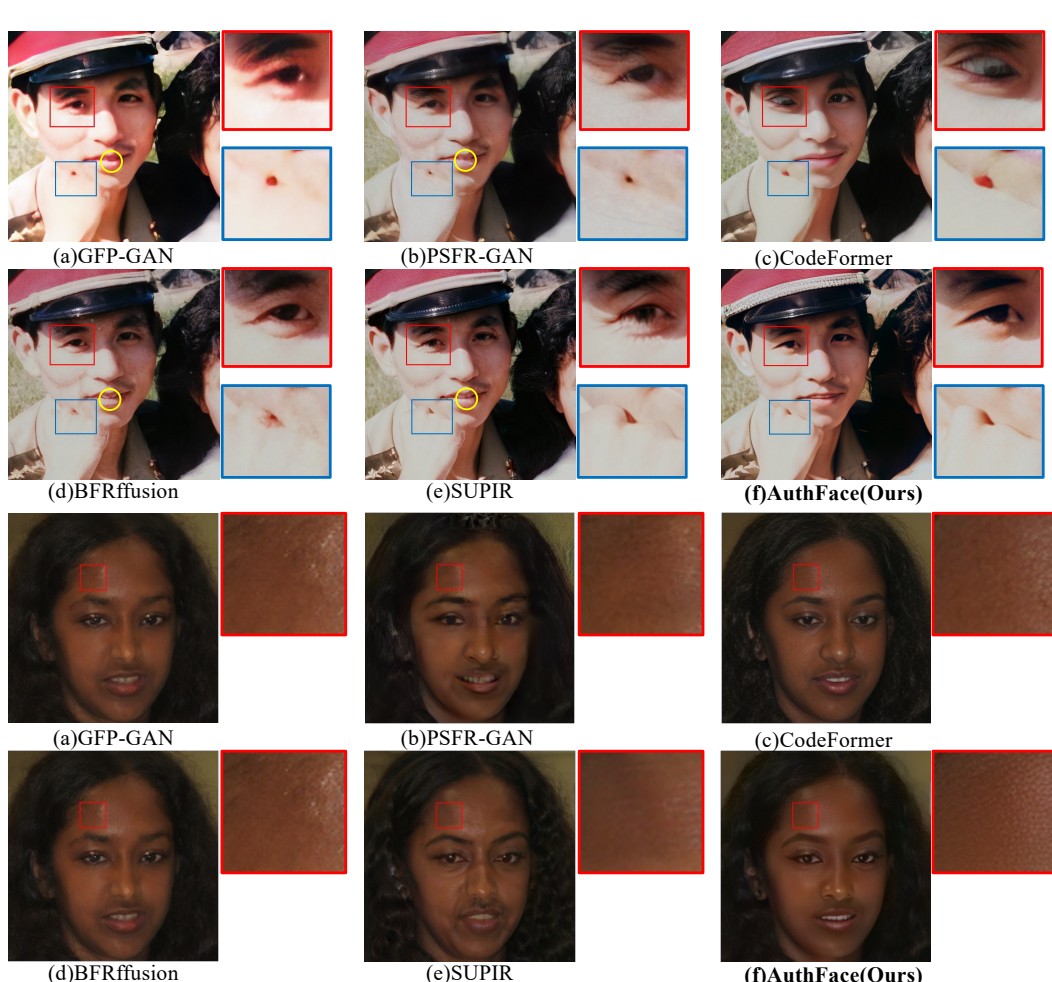

Figure 16: Visualization results in WebPhoto-Test and WIDER-Test dataset. Results in WIDER-Test dataset (2nd case) include a zoomed-in view of the skin highlighted in red box areas. Zoom in for more details.

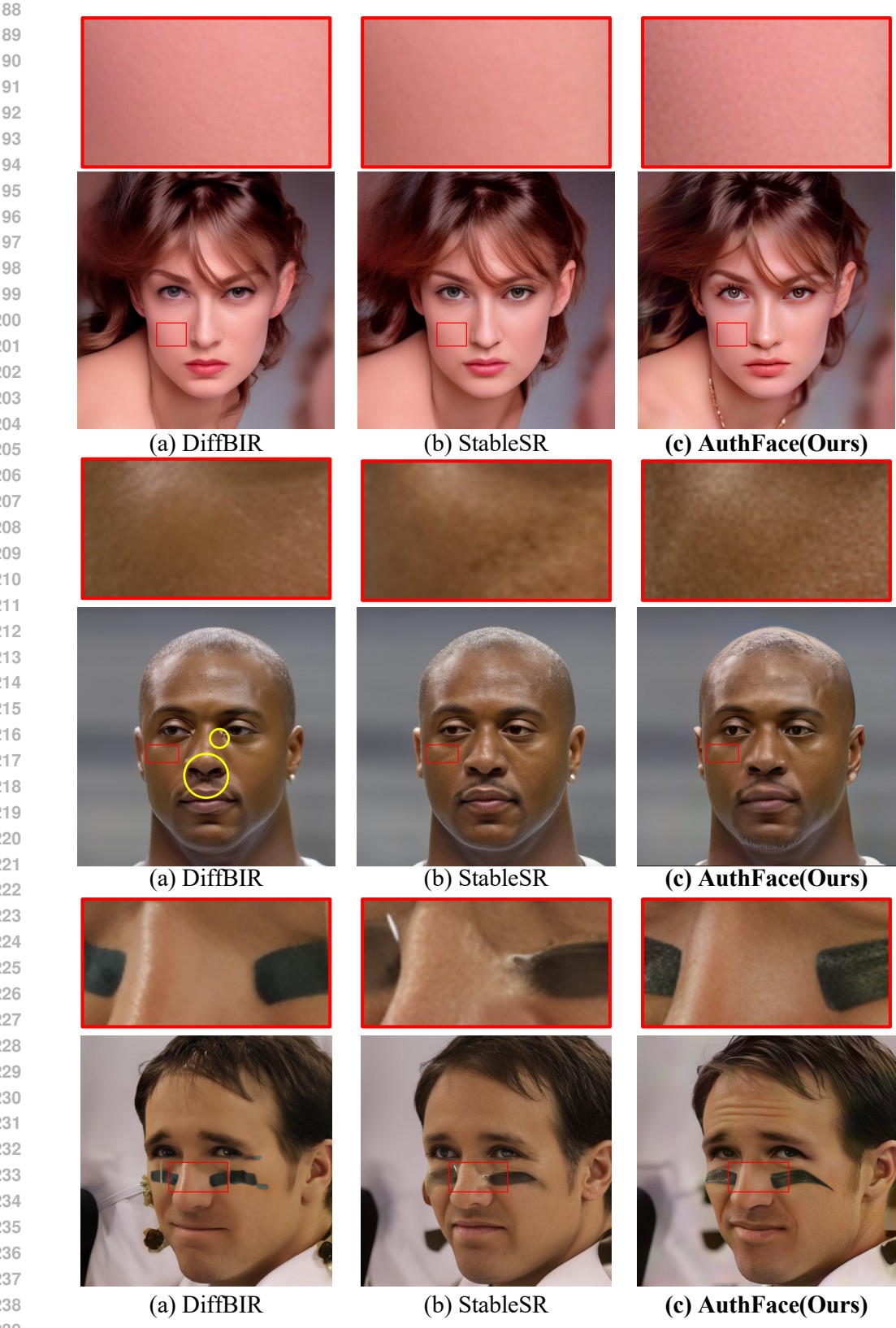

Figure 17: Visualization results in CelebA-Test and LFW-Test dataset including a zoomed-in view of the skin highlighted in red box areas. Zoom in for more details. *Yellow circles in the middle row indicate artifacts that appear around the corners of the eyes and nostrils.*

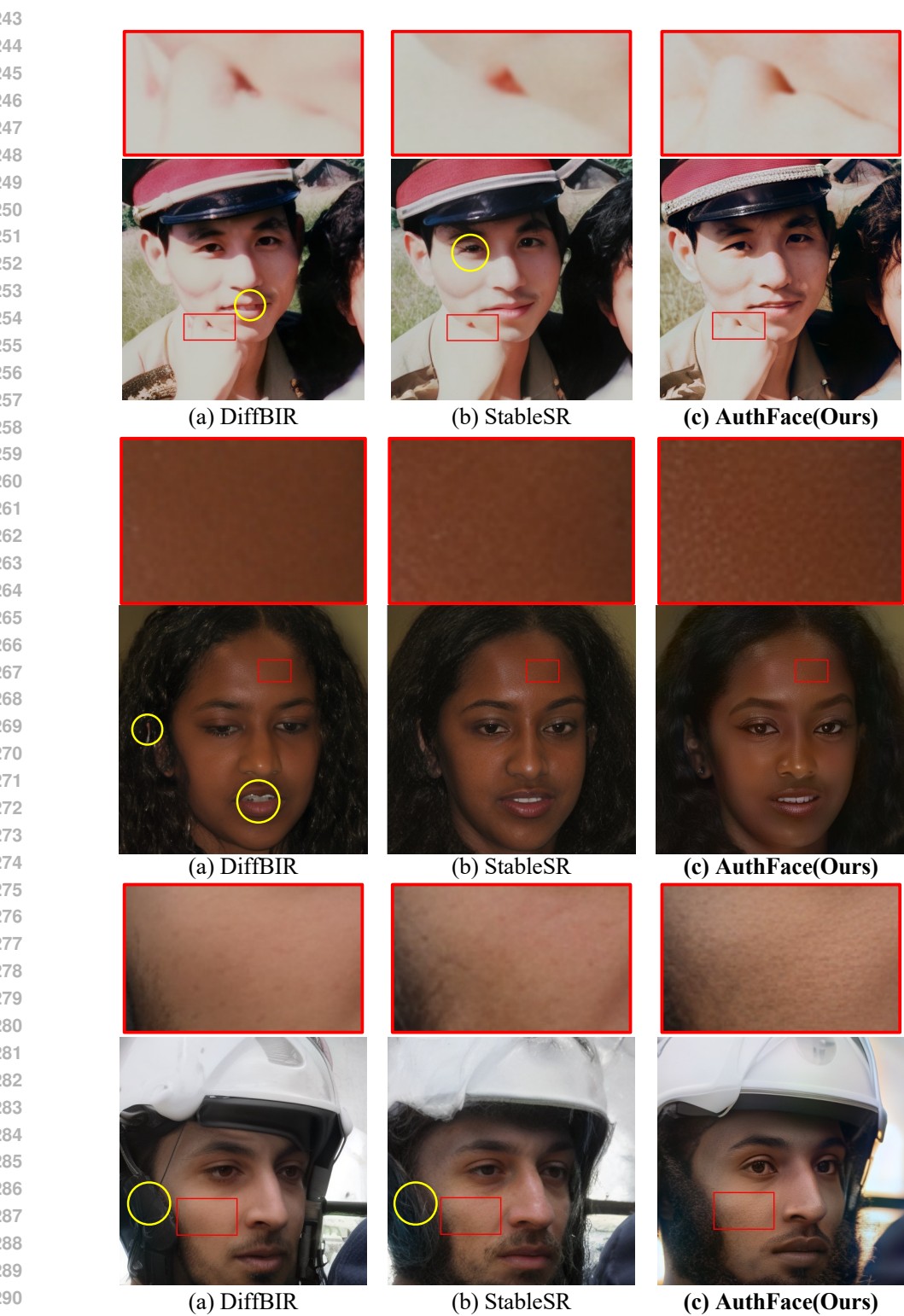

Figure 18: Visualization results in WebPhoto-Test and WIDER-Test dataset including a zoomed-in view of the skin highlighted in red box areas. Zoom in for more details. *Yellow circles in the last row highlight artifacts resulting from the prior's lack of facial detail representation.*

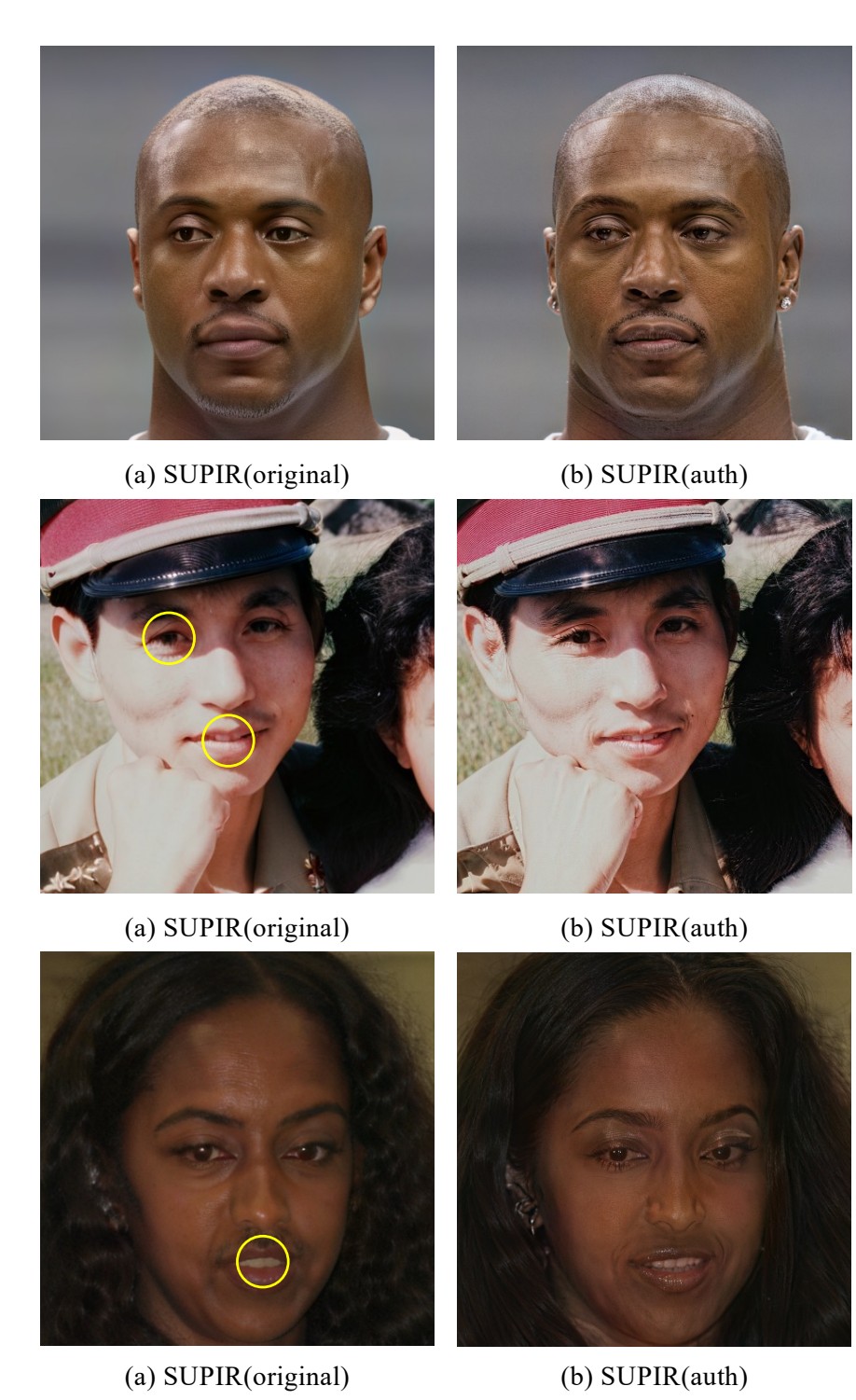

(a) SUPIR(original)    (b) SUPIR(auth)

(a) SUPIR(original)    (b) SUPIR(auth)

(a) SUPIR(original)    (b) SUPIR(auth)

Figure 19: Visualization results of the universal experiment in real-world datasets. Zoom in for more details. *With our face-oriented diffusion prior, SUPIR(auth) improves facial details and avoids incorrect generations.*

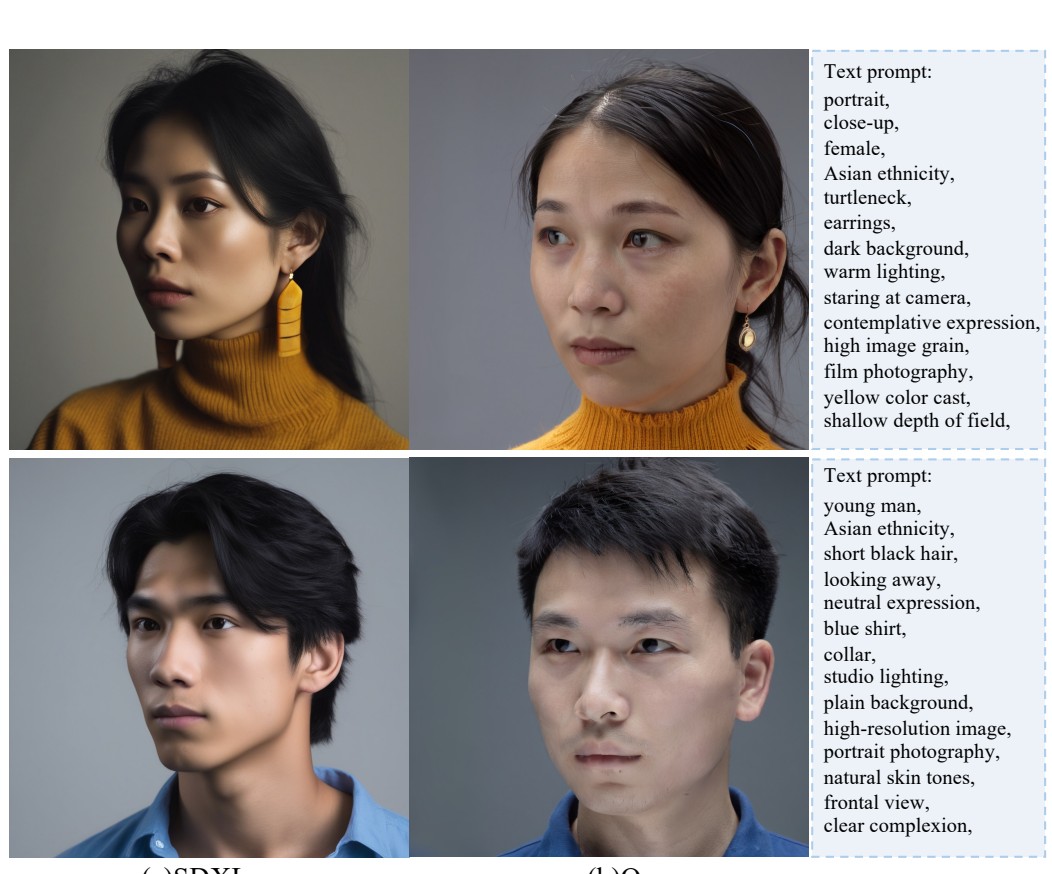

Text prompt:
portrait,
close-up,
female,
Asian ethnicity,
turtleneck,
earrings,
dark background,
warm lighting,
staring at camera,
contemplative expression,
high image grain,
film photography,
yellow color cast,
shallow depth of field,

Text prompt:
young man,
Asian ethnicity,
short black hair,
looking away,
neutral expression,
blue shirt,
collar,
studio lighting,
plain background,
high-resolution image,
portrait photography,
natural skin tones,
frontal view,
clear complexion,

(a)SDXL       (b)Ours

Figure 20: Qualitative comparison between SDXL and our fine-tuned model for the T2I task: SDXL often generates overly smooth skin, whereas our model preserves authentic skin texture. Please zoom in for finer details.

