# OpenReview forum: "AuthFace: Towards Authentic Blind Face Restoration with Face-oriented Generative Diffusion Prior"
_ICLR.cc/2025/Conference — Submitted to ICLR 2025_

### Official Review · Reviewer_pKuC · 2024-10-29

**Soundness:** 2
**Presentation:** 2
**Contribution:** 2
**Rating:** 6
**Confidence:** 4

**Summary:**

The paper proposes a method for blind face restoration. It fine-tunes generative models with high-quality face data and uses a time-aware facial feature loss, improving detail restoration in key areas like eyes and mouth. AuthFace outperforms current methods in both synthetic and real-world tests.

**Strengths:**

1. This paper introduces a high-quality face dataset tailored for restoration tasks, enhancing model training with detailed facial features.
2. This method achieves strong results in restoring realistic facial details, outperforming existing methods in key areas like eyes and mouth across various datasets.

**Weaknesses:**

1. In Figure 2(b), the fine-tuned model shows only marginal improvements over the baseline SDXL model, which brings into question whether the fine-tuning process effectively enhances facial detail restoration.
2. While the study emphasizes the use of photographic prompts during fine-tuning, it lacks specific experiments to verify if these prompts genuinely contribute to improved model performance.
3. The authors did not incorporate photographic prompts in the second stage of training or during testing. Exploring whether adding these prompts could further enhance the model’s restoration quality would be beneficial.
4. Since AuthFace uses a specially collected, high-quality face dataset, the comparison with other methods might not be entirely fair. Competing methods didn’t have access to this dataset, which could make it hard to tell if AuthFace’s improvements come from the method itself or just from having better data.

**Questions:**

See the Weaknesses.

---

### Official Review · Reviewer_FWib · 2024-11-01

**Soundness:** 3
**Presentation:** 3
**Contribution:** 2
**Rating:** 6
**Confidence:** 5

**Summary:**

The paper introduces AuthFace, a framework that achieves highly authentic face restoration by leveraging a face-oriented generative diffusion prior. This work first proposes a dataset of 1.5K HQ face images to fine-tune the pretrained T2I models. Then, the ControlNet is used for training a restoration model for LQ to HQ, where a time-aware latent facial feature loss is proposed to train this restoration model.

**Strengths:**

1. A dataset of 1.5K HQ face images is proposed.
2. The generated faces look good and closer to a real face.

**Weaknesses:**

My concerns are as follows:
1. About the proposed dataset. After a careful reading, I observe that the proposed dataset is only used in stage one, i.e., used to fine-tune the prestrained T2I model, which means that the dataset seems to be solving the problem of generating faces only, not for the BFR task. This can also be seen in Figure 4, where the authors are using an example from the FFHQ dataset to plot.
2. Concerns about methodological innovativeness. The methodological contribution of this work is simply to fine-tune the existing T2I model, i.e., SDXL and ControlNet, and the novelty may be inadequate.
3. For the proposed time-aware latent facial feature loss, why consider only the eyes and mouth? For the facial structures, the nose is very important.

**Questions:**

please see Weaknesses

**Details Of Ethics Concerns:**

This paper contains a new asset, which includes large face images. Thus, the discrimination/bias as well as security/copyright need to be concerns.

---

### Official Review · Reviewer_Bd1a · 2024-11-02

**Soundness:** 3
**Presentation:** 3
**Contribution:** 3
**Rating:** 6
**Confidence:** 5

**Summary:**

This work focuses on the problem of blind face restoration (BFR). The authors aim to overcome the limitations of existing methods using pre-trained text-to-image diffusion models and propose AuthFace for BFR. Specifically, AuthFace leverages a dataset of 1,500 high-quality images taken by professional photographers to create a face-oriented generative diffusion prior. It employs a restoration-tuning pipeline guided by quality-first annotations to enhance facial features and introduces a time-aware latent facial feature loss to minimize artifacts in critical areas. Experimental results demonstrate its superior performance on synthetic and real-world BFR datasets.

**Strengths:**

1. The idea of using a restoration-tuning pipeline guided by quality-first annotations to enhance facial features to guide the  BFR is interesting.
2. Overall, the writing of this paper is good and easy to follow.
3. The experimental results look good on some benchmark datasets.

**Weaknesses:**

1.  While using high-quality face images to fine-tune the pre-trained model is likely important for generating better face priors, the proposed method seems to rely on an additional 1,500 high-quality images, which may be only a trick compared to methods like BFRfusion and SUPIR.
2. In Fig. 6 and Fig. 7, the restored face images exhibit artifacts. For instance, the left eye is closed in the ground truth (GT) image, but in the restoration produced by AuthFace, the eye appears partially open.
3. Some universal evaluations should be conducted. The authors do not use the pre-trained T2I models (fine-tuned after incorporating the 1,500 high-quality images) on existing methods, which could validate the effectiveness of the proposed approach in comparison to those methods.
4. The proposed method seems very heavy. Some complexity results should be provided, such as parameters, FLOPs, and runtime. They can help to more fairly assess the true effectiveness of the method.

**Questions:**

1. Do the authors analyze how the number of high-quality face images enables the pretrained T2I model to produce a more accurate face prior? I think that 1.5K high-quality face images may not be sufficient for effectively fine-tuning the T2I model.
2. Can the authors provide more universal evaluations (refer to the weaknesses section)?

---

### Official Review · Reviewer_jLZ9 · 2024-11-04

**Soundness:** 3
**Presentation:** 2
**Contribution:** 2
**Rating:** 5
**Confidence:** 5

**Summary:**

To tackle the problem of missing details and artifacts, this paper proposes Authface, a novel BFR method with face-oriented generative diffusion prior, designed to restore highly authentic face images. This paper proposes an HQ dataset containing 1.5K images with a resolution exceeding 8K. Based on it, the pretrained T2I model is fine-tuned with semantic and photographic prompts. After tuning, authors leverage ControlNet and a novel time-aware latent facial feature loss to achieve highly authentic face restoration.

**Strengths:**

1. This paper proposes a New High-quality Dataset, with a resolution exceeding 8K.
2. This paper proposes a novel loss to enhance mouth and eye regions, specifically designed for face restoration tasks.
3. Quantitative experiment results show excellent performance for blind face restoration on synthetic and real-world datasets.

**Weaknesses:**

1. The visual image in Figure 1 does not convince me that Authface performs better than SUPIR in the task of missing details and incorrect details. For example, in the red box in the first row, Authface lacks more details than SUPIR.
2. In qualitative experiments, Authface does not perform as well as BFRffusion. For example, in the fourth row of Figure 6, the eyes of BFRffusion are closer to GT than Aythface, and in the first two rows of Figure 7, there is little difference between the two methods. In fact, the image quality of the teeth of BFRffusion is better.
3. More Sota compared methods. Please show more comparisons on more Sota methods, like DiffBIR and StableSR, which also leverage ControlNet as a part of the framework.
4. Contribution. As mentioned above, DiffBIR, StableSR, and so on, have already enhanced the generative capabilities of pretrained T2I models for BFR, which is not a novel research direction.

**Questions:**

1. As mentioned in Weaknesses, more qualitative and quantitative comparisons with Sota methods are needed.
2. It is recommended to replace the image in Figure 1 to make this challenge more convincing.

It was a bit of work to collect this dataset, and I would raise the score if the author could fix the mentioned problems.

---

### Meta-Review · Area_Chair_hwon · 2024-12-19

**Metareview:**

The paper proposes a method for blind face restoration by fine-tuning a pretrained text-to-image (T2I) model on high-quality facial data and subsequently training a ControlNet for face restoration. To support this approach, the authors collected a new high-quality face dataset with resolutions exceeding 8K, along with photographic focus captions.

Strengths
* The paper introduces a high-quality facial dataset with exceptionally large resolution, which could benefit face restoration research.
* The method demonstrates solid quantitative performance on established metrics and benchmarks.

Weaknesses
* The qualitative results are not consistently convincing, with some reviewers noting that the restored faces lack sufficient details or contain artifacts.
* The comparison with state-of-the-art methods is incomplete, and the fairness of the comparisons is questionable due to the additional training data used by the proposed method.
* The technical contribution is limited. The work relies heavily on existing methods that has been well-studied.

Despite the introduction of a high-quality dataset and quantitative improvements, multiple reviewers expressed concerns about the limited technical novelty, unfair comparisons, and less-than-convincing qualitative improvements. The authors’ rebuttal did not fully address these concerns, particularly regarding the fairness of the evaluation and the practical effectiveness of the approach. As a result, the paper's overall contribution is considered insufficient to warrant acceptance.

**Additional Comments On Reviewer Discussion:**

1. The authors attempted to address concerns about the non-convincing qualitative results, but the provided clarifications did not fully resolve this issue.
2. Additional experiments and comparisons were added to address concerns about limited SOTA comparisons, effectively resolving this concern.
3. While the authors provided arguments to justify the fairness of their evaluation, these did not fully convince the reviewers.
4. The authors pointed to relevant experiments to justify the use of photography-guided prompts, successfully resolving concerns about their insufficient justification.

---

### Decision · Program_Chairs · 2025-01-22

Reject